# Peer review of "The Pharmacology of Xenobiotics after Intracerebro Spinal Fluid Administration: Implications for the Treatment of Brain Tumors"

_ijms, 2021, doi:10.3390/ijms22031281_

Round 1
Reviewer 1 Report
The topic of the work of Paris et al. addresses a clinical very relevant and relatively understudied topic. Thus, the review certainly qualifies for publication with high priority based from the topic. However, the content is insufficient in the current form and requires severe edits. If you take out the - to my taste - mostly irrelevant section 2 (see below), the paper is relatively short and concise for a normal review, rather a mini-review.
Main points:
Section 2 is irrelevant for the topic. The minimal summary – presenting basic topics - on the neuroanatomy provided in the article is not fair for the complex research field. Obviously, much more is known. I suggest fusing the main points/problems of this section in the intro.
General routes regarding the activities on how promising substances for intrathecal application to treat cerebal mets/ brain tumors in the context of personalized medicine shall be discussed. I refer to screening approaches in vitro using primary tumor material or there-of derived cell models.
L 238 The listing is incomplete. General aspects of “competitors” of intrathecal drug shall be discussed. I refer to other surgical-assisted intra-cerebral brain delivery such a through intra-arterial catheter-mediated injection; or improving BBB crossing of substances via nano-functionalization of drugs. After that, the main benefit/advantage – and possible downsides- of intrathecal injections from patient standpoint, costs, clinical readiness level, dissemination potential etc. compared to the main competitor approaches could highlight the relevance to intrathecal delivery route. This will give the article more prominent position to a wider audience.
Wrongful statement on prevalence of CNS metastasis ??
"The exact incidence of brain metastases, which develop in nearly 30% of patients with solid tumours (5.), is unknown but is rising given the longer survival durations and better prognosis of many patients with cancer, the use of surveillance imaging, an ageing population, and greater levels of awareness among oncologists. Data from autopsy cohorts indicate that the true incidence of brain metastases is higher among patients with certain primary cancers (6., 7.).“
(From: Suh JH, Kotecha R, Chao ST, Ahluwalia MS, Sahgal A, Chang EL. Current approaches to the management of brain metastases. Nat Rev Clin Oncol. 2020 May;17(5):279-299. doi: 10.1038/s41571-019-0320-3. Epub 2020 Feb 20. PMID: 32080373.)
- Scoccianti, S. & Ricardi, U. Treatment of brain metastases: review of phase III randomized controlled trials. Radiother. Oncol. 102, 168–179 (2012).
- Tsukada, Y., Fouad, A., Pickren, J. W. & Lane, W. W. Central nervous system metastasis from breast carcinoma. Autopsy study. Cancer 52, 2349–2354 (1983).
- Sampson, J. H., Carter, J. H. Jr., Friedman, A. H. & Seigler, H. F. Demographics, prognosis, and therapy in 702 patients with brain metastases from malignant melanoma. J. Neurosurg. 88, 11–20 (1998).
The main reference of this article are the tables. Hwoever, they are not complete, i.e.:
Table 1: Add anti-cancer drugs: TKIs like Gefitinib, Afatinib and Osimertinib:
Ballard P, Yates JW, Yang Z, Kim DW, Yang JC, Cantarini M, Pickup K, Jordan A, Hickey M, Grist M, Box M, Johnström P, Varnäs K, Malmquist J, Thress KS, Jänne PA, Cross D. Preclinical Comparison of Osimertinib with Other EGFR-TKIs in EGFR-Mutant NSCLC Brain Metastases Models, and Early Evidence of Clinical Brain Metastases Activity. Clin Cancer Res. 2016 Oct 15;22(20):5130-5140. doi: 10.1158/1078-0432.CCR-16-0399. Epub 2016 Jul 19. PMID: 27435396.
Soria JC, Ohe Y, Vansteenkiste J, Reungwetwattana T, Chewaskulyong B, Lee KH, Dechaphunkul A, Imamura F, Nogami N, Kurata T, Okamoto I, Zhou C, Cho BC, Cheng Y, Cho EK, Voon PJ, Planchard D, Su WC, Gray JE, Lee SM, Hodge R, Marotti M, Rukazenkov Y, Ramalingam SS; FLAURA Investigators. Osimertinib in Untreated EGFR-Mutated Advanced Non-Small-Cell Lung Cancer. N Engl J Med. 2018 Jan 11;378(2):113-125. doi: 10.1056/NEJMoa1713137. Epub 2017 Nov 18. PMID: 29151359.
Table 2: Add study with Elacridar and Gefitinib:
Agarwal S, Sane R, Gallardo JL, Ohlfest JR, Elmquist WF. Distribution of gefitinib to the brain is limited by P-glycoprotein (ABCB1) and breast cancer resistance protein (ABCG2)-mediated active efflux. J Pharmacol Exp Ther. 2010 Jul;334(1):147-55. doi: 10.1124/jpet.110.167601. Epub 2010 Apr 26. PMID: 20421331; PMCID: PMC2912048
- 202: There is more (and recent) data about Pgp-inhibitors in humans:
Elkhayat HA, Aly RH, Elagouza IA, El-Kabarity RH, Galal YI. Role of P-glycoprotein inhibitors in children with drug-resistant epilepsy. Acta Neurol Scand. 2017 Dec;136(6):639-644. doi: 10.1111/ane.12778. Epub 2017 May 31. PMID: 28560774.
- 239-240: Add approach of disruption of BBB by (whole brain) radiation:
Zeng YD, Liao H, Qin T, Zhang L, Wei WD, Liang JZ, Xu F, Dinglin XX, Ma SX, Chen LK. Blood-brain barrier permeability of gefitinib in patients with brain metastases from non-small-cell lung cancer before and during whole brain radiation therapy. Oncotarget. 2015 Apr 10;6(10):8366-76. doi: 10.18632/oncotarget.3187. PMID: 25788260; PMCID: PMC4480758.
Pollack IF, Stewart CF, Kocak M, Poussaint TY, Broniscer A, Banerjee A, Douglas JG, Kun LE, Boyett JM, Geyer JR. A phase II study of gefitinib and irradiation in children with newly diagnosed brainstem gliomas: a report from the Pediatric Brain Tumor Consortium. Neuro Oncol. 2011 Mar;13(3):290-7. doi: 10.1093/neuonc/noq199. Epub 2011 Feb 3. PMID: 21292687; PMCID: PMC3064605.
Minor points:
Are there trends of increased infections/ complications for cancer patient’s receiving intrathecal injections or other common unwanted effects/risks?
L 40: have a limited benefit on cancer brain metastases … this is not true to many efforts, or at least not known (in case of immune therapy). Shall be de-toned a bit to make a lighter statement.
Appreciation of current results on p-glycoprotein inhibitor in glioma management is missing: https://www.ncbi.nlm.nih.gov/pmc/articles/PMC6628362/
Reviewer 2 Report
In this review submitted by Paris et al., the authors enlightened the pharmacology of xenobiotics after intra-thecal spinal fluid administrations and its implications for the treatment of brain tumors. Brain metastasis has been a long issue for the cancer patients. Targeting brain tumors are specifically difficult due to complicated structure and barriers called Blood-Brain Barriers. In the recent times, due to technical advancements in the field, this issue has been addressed but how this drugs are metabolized and excreted, very little is known. In this review, the authors have discussed the implications of overcoming this barriers and drugs that are used across this barriers.
The authors divided this manuscript in four divisions. Introduction, Anatomical structures, Turnover in the glymphatic system followed by Efflux systems and drug clearance. First two sections are very clearly written and easy to follow up. Glymphatic system and efflux system should be re written in a clear manner. The flow of those section is not good so I suggest that those section needs to be rewritten in a way that is easy to follow for the readers. Also, the figure-2 requires a slight work. This color coded molecules going through the transporters needs to be shown as what those molecules are. This can be done by writing it in figure legends.
Round 2
Reviewer 1 Report
the authros did ok job for the revision.
Reviewer 2 Report
To my satisfaction, the authors have addressed the concerns raised in the first review. Thus, the review can now be accepted in its current form.